# Influence of Particle Size and Extraction Methods on Phenolic Content and Biological Activities of Pear Pomace

**DOI:** 10.3390/foods12234325

**Published:** 2023-11-29

**Authors:** Joana Ferreira, Karolina Tkacz, Igor Piotr Turkiewicz, Maria Isabel Santos, Adriana Belas, Ana Lima, Aneta Wojdyło, Isabel Sousa

**Affiliations:** 1LEAF—Linking Landscape, Environment, Agriculture and Food Research Center, Associate Laboratory TERRA, Instituto Superior de Agronomia, Universidade de Lisboa, Tapada da Ajuda, 1349-017 Lisboa, Portugal; missantos@gmail.com (M.I.S.); agusmaolima@gmail.com (A.L.); isabelsousa@isa.ulisboa.pt (I.S.); 2Department of Fruit, Vegetable and Plant Nutraceutical Technology, Faculty of Biotechnology and Food Science, Wrocław University of Environmental and Life Sciences, 37 Chełmońskiego Street, 51-630 Wrocław, Poland; karolina.tkacz@upwr.edu.pl (K.T.); igor.turkiewicz@upwr.edu.pl (I.P.T.); aneta.wojdylo@upwr.edu.pl (A.W.); 3Veterinary and Animal Research Centre (CECAV), Faculty of Veterinary Medicine, Lusófona University, 376 Campo Grande, 1749-024 Lisboa, Portugal; p6226@ulusofona.pt

**Keywords:** pear pomace, antidiabetic, ACE, TPC, TFC, antioxidant

## Abstract

The main goal of this research was to investigate how particle size influences the characteristics of pear (*Pyrus Communis* L.) pomace flour and to examine the impact of different pre-treatment methods on the phenolic content and associated bioactivities. Pear pomace flour was fractionated into different particle sizes, namely 1 mm, 710 µm, 180 µm, 75 µm and 53 µm. Then two extraction methods, namely maceration with methanol and two-step extraction with hexane via Soxhlet followed by ultrasound extraction with methanol, were tested. Total phenolic and total flavonoid contents ranged from 375.0 to 512.9 mg gallic acid/100 g DW and from 24.7 to 34.6 mg quercetin/100 g DW, respectively. Two-step extraction provided antioxidant activity up to 418.8 (in FRAP assay) and 340.0 mg Trolox/100 g DW (in DPPH assay). In order to explore various bioactive properties, this study assessed the inhibitory effects of enzymes, specifically α-amylase and β-glucosidase (associated with antidiabetic effects), as well as angiotensin-converting enzyme (linked to potential antihypertensive benefits). Additionally, the research investigated antibacterial potential against both Gram-negative (*E. coli*) and Gram-positive (*S. aureus*) bacteria, revealing significant results (*p* < 0.05), particularly in the case of the two-step extraction method. This investigation underscores the substantial value of certain food industry wastes, highlighting their potential as bioactive ingredients within the framework of a circular economy.

## 1. Introduction

Pear (*Pyrus communis* L.) belongs to the rose family (Rosaceae) and is one of the oldest and most commonly cultivated plants. There are several thousand cultivars of pears in the world, of which about 100 are commercially grown [1]. Overall, the global production of pears in 2021 reached over 26 million tonnes, of which 80.1% was from Asia and 10.2% was from European countries [2].

There are several processing techniques to produce pear juice; the pear juice pressing process is the most commonly used. This juice extraction method produces 35% pomace with good nutritional quality as waste, which causes waste management, economic and environmental issues [3]. Pear pomace is highly perishable due to being 60–70% moisture and has been traditionally used as an additive to livestock feed or is wasted [4,5]. However, reintroducing fruit pomace back into the food value chain, is more sustainable and environmentally friendly and increases fruit processing industry efficiency. In addition, the market for fruit pomace use was valued at $3.2 billion in 2020, and CAGR is forecasted to grow 6.8% by 2026, due to the increase in consumers’ awareness of the importance of sustainability and healthier eating and the growing interest in natural and organic food products [6]. Whilst the production of pear-based products continues to increase, the importance of optimizing strategies for pear pomace management and for its promotion in a context of green production and value-added products.

Pear pomace is available in various forms, including wet pomace (in paste form), flour and pellets. In the food industry, pear pomace finds its primary applications in dairy products, beverage processing, the extraction of aroma and flavor compounds, pectin production, additives to cereals and enhancers of bread quality [6]. In addition to these applications, pear pomace can serve various other purposes, such as animal feed and potential cosmetic and nutraceutical applications of its extracts, although the latter are less common.

From a nutritional standpoint, pear pomace is recognized as a valuable source of dietary fiber, which comprising a substantial 90.7%, with a soluble fraction of 1.5%, which can contribute to reducing the risk of diet-related diseases. It also contains minerals (1% as ash), minimal free sugars (0.3%), a modest amount of fat (3.7%) and a protein content of 5.7% [4]. Recent research on the promotion of fruit and vegetable by-products has led to the development of extraction methods for recovering lipids, proteins, phenolics and fiber from such waste materials [7]. Pear pomace, with its constituents of seeds, peel and pulp, contains a complex of biologically active compounds [1,8]. In fact, pear consumption has been associated with a multitude of health-promoting effects, which have been extensively documented [1].

The high concentration of bioactive compounds in pear fruit has been associated with antioxidant, anti-obesity (as pancreatic lipase inhibition), anti-aging (as cholinesterase inhibition) and anti-inflammatory (as COX-1, COX-2, 15-LOX inhibition) activities. It has been shown to reduce the incidence of strokes and lung cancer, as well as anti-ulcer activity, after supplementation with pear-enriched formulas [1,8]. A number of studies have emphasized the effectiveness of regular consumption of pears in lowering blood pressure and reducing the risks of type 2 diabetes, which is additionally supported by their low caloric value (57 kcal/100 g) and their low (IG = 38) glycemic index [1]. In fact, Cisneros-Yupanqui and co-workers (2023) reported that pears are remarkable natural sources of bioactive compounds that constitute an alternative to the synthetic drugs used to control noncommunicable diseases such as diabetes, in particular type 2 (T2DM), by inhibiting digestive enzymes, such as α-amylase and α-glucosidase [9]. Nevertheless, all of these reports relate to fresh raw fruit, and literature data on the health potential of pear pomace is limited. However, there is a possibility that pear pomace yields a higher concentration of these bioactives, but these applications have been underexplored.

One of the commonly used processing methods is convection drying, ensuring microbiological stability and increasing the storage life of the dry product [5]. The trade-off between quality and cost-effectiveness is a major challenge for food producers. However, the physical properties, including the particle size of the dried pomace, should be considered in terms of phytochemical composition and potential bioactivity-promoting properties.

Our research is centered on exploring a specific waste product from the juice industry, namely pear pomace, which has been relatively overlooked as a source of bioactive antioxidant compounds. This study presents a collection of findings regarding the health-promoting potential of pear pomace and investigates how physicochemical parameters, such as particle size and the method of bioactive compound extraction, can influence its biological activity. In this context, our main goals were to assess the potential bioactivities of pear pomace and identify the most effective processing methods to enhance them. To achieve this, we examined the impact of granulation level and particle size of pear pomace powder, along with the pre-treatment method, on the phenolic content and various bioactivities of pomace, including antioxidant, antidiabetic, antihypertensive and antibacterial properties.

Collectively, our findings offer insights into optimizing the production of pear powder by selecting the appropriate grinding level and specific granulation fractions to maximize their bioactive potential, while promoting environmentally friendly production practices. This study thus provides fresh perspectives on the utilization of pear pomace powder in the development of food products with enhanced health benefits and nutraceutical applications, with a particular focus on its potential to reduce blood sugar levels and contribute to cardiovascular health.

## 2. Materials and Methods

### 2.1. Reagents and Standards

Ultrapure water was obtained from Synergy^®^ Water Purification System (Merck Millipore, Burlington, VT, USA). α-amylase (thermostable, from Bacillus sp., 3000 U/mL, E-BSTAA) was purchased from Megazyme (Wicklow, Ireland). β-glucosidase (from almonds, ≥2 units/mg solid, G0395-2.5KU) and Angiotensin Converting Enzyme (ACE, from rabbit lung, ≥2.0 units/mg protein, A6778-0.25UN) were purchased from MilliporeSigma (Saint Louis, MO, USA).

Methanol, hexane, DPPH (2,2-diphenyl-1-picryl-hydrazyl-hydrate), Trolox (6-hydroxy-2,5,7,8-tetramethylchroman-2-carboxylic acid), acetic acid, sodium acetate, TPTZ (2,4,6-tris(2-pyridyl)-s-triazine), hydrochloric acid, iron (III) chloride hexahydrate, Folin–Ciocalteu reagent, sodium carbonate, gallic acid, sodium nitrate (III), aluminum chloride hexahydrate, sodium hydroxide, quercetin dihydrate, DNSA (3,5-dinitrosalicylic acid), potassium sodium tartrate tetrahydrate, sodium phosphate monobasic monohydrate, sodium chloride, soluble starch from potato, maltose monohydrate, sodium citrate, citric acid, p-NPG (4-nitrophenyl-β-D-glucopyranoside), Tris-HCl, FAPGG (N-[3-(2-Furyl)acryloyl]-phenylalanyl-glycyl-glycine) and lisinopril were purchased from MilliporeSigma (Saint Louis, MO, USA). Brain Heart Infusion Broth (BHI), Tryptone Soya Agar (TSA) were purchased from bioMérieux^®^ SA (Marcy l’Étoile, France) and Müller–Hinton medium was purchased from Biokar Diagnostics (Beauvais, France).

### 2.2. Plant Material and Sample Preparation

The research material consisted of pear pomace powder submitted to the process of tunnel-drying (Tecnofruta, Valencia, Spain) under certain conditions (80–85 °C, 110 min, 55 Hz of air flow) and grinding (Ferneto, Vagos, Portugal), obtained from the company ALITEC—Alimentos Tecnológicos SA (Nazaré, Portugal).

The average laboratory sample was sieved through consecutive sieves with the following mesh sizes: 1 mm (2.1%), 710 µm (14.7%), 180 µm (55.0%), 75 µm (20.3%), 53 µm (6.7%). Each fraction was then subjected to different extraction processes. One batch of powder of differing granulations was extracted with 80% (v/v) methanol for 30 min with an ultrasonic bath (Transsonic 700, Elma Schmidbauer GmbH, Singen, Germany) at room temperature, followed by 4 h of shaking in laboratory rotator (Reax 2, Heidolph Instruments GmbH & Co. KG, Schwabach, Germany). A sample-to-solvent ratio of 1:5 (w/v) was used. After this time, the samples were centrifuged for 10 min at 24,104× *g* using a bench cooling centrifuge (Z 383 K, Hermle Labortechnik GmbH, Wehingen, Germany), and the supernatant was filtered through qualitative filter papers (grade 4, Whatman™, Maidstone, UK) and immediately analyzed.

The second batch of powder, with differing granulations, was subjected to the process of purification with hexane using the Soxhlet method (twice for 4 h). The purified samples were dried at 60 °C for 2 h (WTC Binder Climate Chamber Incubator 9010-0021, BINDER GmbH, Tuttlingen, Germany), then they were extracted in the same way as the first batch.

Samples were lyophilized prior to analysis.

### 2.3. Total Phenolic Content (TPC)

The analysis was performed according to the procedure previously described, e.g., by Graça et al. (2020), with modifications [10]: 0.1 mL of extract, 4 mL of Folin–Ciocalteu reagent (diluted 1:10 with water, v/v) and 4 mL of 7.5% (w/v) aqueous sodium carbonate solution were added to a 15 mL plastic test tube with a lid screwed on, vortexed for 30 s (RSLAB-6PRO, Normax, Marinha Grande, Portugal) and incubated for 15 min in a circulating water bath (Precision™ 2864, Thermo Scientific, Waltham, MA, USA) at a temperature of 45 °C while protected against light. After this time, the absorbance was measured via a UV-visible spectrophotometer (Cary 100, Agilent Technologies, Santa Clara, CA, USA) in a 1 cm-thick quartz cuvette at a wavelength of 765 nm against water. For the blank sample, 0.1 mL of water was added instead of the extract. The standard curve was prepared from a water solution of gallic acid at concentrations of 0.06; 0.15; 0.3; 0.4; 0.54 and 0.6 mg/mL (R^2^ = 0.9917). The samples were analyzed in triplicate, and the results were expressed as mg of gallic acid per 100 g DW (dry weight).

### 2.4. Total Flavonoid Content (TFC)

The analysis was performed according to the procedure previously described, e.g., by Queirós et al. (2020), with modifications [11]: 2 mL of extract and 0.12 mL of 5% (w/v) sodium nitrate (III) aqueous solution were placed in a 15 mL plastic tube, mixed and left for 5 min in the dark. Then 0.12 mL of 10% (m/v) aqueous aluminum chloride hexahydrate solution was added, and the mixture was stirred and left in the dark for another 6 min. After this time, 0.8 mL of 4% (m/v) aqueous sodium hydroxide solution was added; after 2 min, 0.96 mL of water was added. The tubes were screwed closed and vortexed for 30 s (RSLAB-6PRO, Normax, Marinha Grande, Portugal). The absorbance was measured with a UV-visible spectrophotometer (Cary 100, Agilent Technologies, Santa Clara, CA, USA) in a 1 cm-thick quartz cuvette at a wavelength of 420 nm against water. For the blank sample, 0.1 mL of water was added instead of the extract. The standard curve was prepared from a methanolic solution of quercetin at concentrations of 0.01; 0.02; 0.04; 0.06; 0.08 and 0.1 mg/mL (R^2^ = 0.9929). The samples were analyzed in triplicate, and the results were expressed as mg of quercetin per 100 g DW.

### 2.5. Antioxidant Analysis

#### 2.5.1. FRAP Assay

The analysis was performed according to the procedure previously described, e.g., by Nunes et al. (2020), with modifications [12]: FRAP (ferric reducing antioxidant power) test solution was prepared by mixing at a ratio of 10:1:1 (v/v/v) 0.3 M acetate buffer at pH 3.6, 10 mM TPTZ (2,4,6-Tris(2-pyridyl)-s-triazine) prepared in 40 mM HCl solution and 20 mM aqueous iron (III) chloride hexahydrate solution. A 0.09 mL quantity of extract, 0.27 mL of water and 2.7 mL of test solution were placed in a 15 mL plastic test tube which was screwed closed, vortexed for 30 s (RSLAB-6PRO, Normax, Marinha Grande, Portugal) and incubated for 30 min in a water bath (Precision™ 2864, Thermo Scientific, Waltham, MA, USA) at a temperature of 37 °C, protected against light. After this time, the absorbance was measured using a UV-visible spectrophotometer (Cary 100, Agilent Technologies, Santa Clara, CA, USA) in a 1 cm-thick quartz cuvette at a wavelength of 593 nm against test solution. The standard curve was prepared from a methanolic solution of Trolox with concentrations at 0.02; 0.05; 0.1 and 0.2 mg/mL, respectively (R^2^ = 0.9942). The samples were analyzed in triplicate, and the results were expressed as mg of Trolox per 100 g DW.

#### 2.5.2. DPPH Assay

The analysis was performed according to the procedure previously described, e.g., by Assunção et al. (2017), with modifications [13]: 3.9 mL of 0.24 mg/mL DPPH (2,2-diphenyl-1-picryl-hydrazyl-hydrate) methanolic solution and 0.1 mL of extract were placed in a 15 mL plastic tube which was screwed closed, vortexed for 30 s (RSLAB-6PRO, Normax, Marinha Grande, Portugal) and incubated for 45 min in a circulating water bath (Precision™ 2864, Thermo Scientific, Waltham, MA, USA) at a temperature of 30 °C, protected against light. After this time, the absorbance was measured using a UV-visible spectrophotometer (Cary 100, Agilent Technologies, Santa Clara, CA, USA) in a 1 cm-thick quartz cuvette at a wavelength of 515 nm against methanol. For the blank sample, 0.1 mL of methanol was added instead of the extract. The standard curve was prepared from a methanolic solution of Trolox (6-hydroxy-2,5,7,8-tetramethylchroman-2-carboxylic acid) with concentrations at 0.02; 0.05; 0.1 and 0.2 mg/mL (R^2^ = 0.9948). The samples were analyzed in triplicate, and the results were expressed as mg of Trolox per 100 g dry weight (DW).

### 2.6. Antidiabetic Analysis

#### 2.6.1. α-Amylase Inhibition Test

The analysis was performed according to the procedure previously described, e.g., by Kifle et al. (2020) and Wickramaratne et al. (2016), with modifications [14,15]: a DNSA (3,5-Dinitrosalicylic acid) color solution was prepared as follows: 12 g potassium sodium tartrate tetrahydrate dissolved while hot (50–70 °C) in 8 mL of 2 M aqueous sodium hydroxide solution. Then 438 mg of 3,5-dinitrosalicylic acid was dissolved in 20 mL of water while hot. To the previously prepared 5.3 M alkaline potassium sodium tartrate tetrahydrate solution, 12 mL of 60 °C water and 96 mM aqueous DNSA solution were added. A 20 mM sodium phosphate buffer containing 6.7 mM sodium chloride at pH 6.9 was prepared by dissolving 0.24 g sodium phosphate monobasic monohydrate and 39 μg sodium chloride in 1 L of water.

#### 2.6.2. β-Glucosidase Inhibition Test

The analysis was performed according to the procedure previously described by Pistia-Brueggeman and Hollingsworth (2001) with modifications [16]: a 0.3 M citrate buffer (0.3 M sodium citrate–0.3 M citric acid) at pH 5.0 was prepared by dissolving 88.2 g sodium citrate in 1 L of water. The pH was then brought to the desired value via addition of an aqueous citric acid solution (63 g/L water). A 0.6 mL quantity of extract, 1.5 mL of buffer and 0.3 mL of aqueous β-glucosidase (0.66 mg/mL) were placed in a 15 mL test tube, which was screwed closed, vortexed for 30 s (RSLAB-6PRO, Normax, Marinha Grande, Portugal) and incubated for 15 min in a circulating water bath (Precision™ 2864, Thermo Scientific, Waltham, MA, USA) at a temperature of 37 °C, protected against light. Then 0.6 mL of 5 mM p-NPG (4-Nitrophenyl-β-D-glucopyranoside) solution (in buffer) was added, and the solution was mixed and incubated for another 20 min. After this time, 1.5 mL of 0.1 M sodium carbonate aqueous solution was mixed in. The absorbance was measured using a UV-visible spectrophotometer (Cary 100, Agilent Technologies, Santa Clara, CA, USA) in a 1 cm-thick quartz cuvette at a wavelength of 405 nm against water. For the blank sample, buffer was added instead of the extract. The samples were analyzed in triplicate, and the results were expressed as % of inhibition using the following formula: % relative inhibition: [(Blank-Sample)/Blank] × 100%. The results were given for the sample quantity of 48.10 mg for 1 mL of enzyme.

### 2.7. Antihypertensive Analysis as ACE (Angiotensin-Converting Enzyme) Inhibition Test

The analysis was performed according to the procedure previously described by Murray et al. (2004), with modifications [17]. A 50 mM Tris-HCl buffer containing 300 mM NaCl and 0.1 M HCl at pH 8.3 was prepared by dissolving 7.88 g of Tris-HCl base and 17.532 g of sodium chloride in 1 L of 0.1 M aqueous hydrochloric acid. A 5 M aqueous sodium hydroxide solution was used to adjust the pH. A 0.5 mL quantity of 1.6 mM FAPGG (N-[3-(2-Furyl)acryloyl]-phenylalanyl-glycyl-glycine) solution (in buffer), 0.8 mL of buffer and 0.1 mL of extract were placed in a 15 mL plastic tube, which was screwed closed, vortexed for 30 s (RSLAB-6PRO, Normax, Marinha Grande, Portugal) and incubated for 10 min in a circulating water bath (Precision™ 2864, Thermo Scientific, Waltham, MA, USA) at a temperature of 37 °C, protected against light. Then 0.4 mL of an aqueous ACE solution (0.05 U/mL) was added and mixed in. The decrease in absorbance was measured using a UV-visible spectrophotometer (Cary 100, Agilent Technologies, Santa Clara, CA, USA) in a 1 cm-thick quartz cuvette at a wavelength of 340 nm against water for a period of 60 min. Measurements were taken every 5 min. For the blank sample, buffer was added instead of the extract. The absolute value of the decrease in absorbance per minute (ΔAbs/min) was used for the calculation. For positive control (100% inhibition), a solution of lisinopril with a concentration of 5 µM in assay solution was used. The samples were analyzed in triplicate, and the results were expressed as % of inhibition using the following formula: % relative inhibition: [(Blank-Sample)/Blank] × 100%. The results were given for the sample quantity of 6.01 mg for 1 mL of enzyme.

### 2.8. Determination of Antibacterial Activities

The Gram-positive bacteria species used to assess antibacterial activity of pear pomace extracts was *Staphylococcus aureus* ATCC 6538 (Instituto Nacional de Saúde Dr. Ricardo Jorge—INSA, Lisboa, Portugal). The Gram-negative strain tested was *Escherichia coli* O157:H7 NCTC 12900 (verotoxin negative). Strains were recovered from a culture at −80 °C and placed into 10 mL of Brain Heart Infusion Broth (BHI) (Oxoid, Bansingstone, UK) for two consecutive cultures at 24 h intervals, inoculated afterwards on Tryptone Soya Agar (TSA) (bioMérieux^®^ SA, Marcy l’Étoile, France) and incubated at 37 °C ± 1 °C overnight. The antibacterial activity of each sample was determined as described by Habiba et al. (2015) [18]. In short, 50 μL of Müller–Hinton medium (Biokar Diagnostics, Beauvais, France) was added to each well, and 50 µL of each sample, diluted in fresh media to reach a final concentration of 5.0 mg/mL in each well, was added. Subsequently, 50 µL of the bacterial suspension at a concentration of 2 × 105 CFU/mL was added to the wells. A positive and a negative control, 50 µL of Müller-Hinton medium + 50 µL bacterial suspension and 100 µL of Müller-Hinton medium, respectively, were also performed. The plates were incubated for 24 h at 37 °C, and the absorbance readings were taken at 546 nm (Synergy HT, Biotek, Winooski, VT, USA).

### 2.9. Statistical Analysis

The one-way analysis of variance (ANOVA), Tukey’s HSD test, Tukey’s multiple comparison test, Pearson’s correlation coefficients, agglomerative hierarchical clustering (AHC) and principal component analysis (PCA) were applied using Origin Statistical Software for Excel version 2021.4.1 (Addinsoft, New York, NY, USA) integrated with Microsoft Excel 2021 (Microsoft Corp., Redmond, WA, USA). A level of *p* ≤ 0.05 was considered significant. Scatter plots were developed using Zenplot 1.0.8. by Addinsoft.

## 3. Results

### 3.1. Total Phenolic Content and Total Flavonoid Content

Phenolic compounds are among the most important contributors to the antioxidant activity of fruits and vegetables. The spectrophotometric method with Folin–Ciocalteu reagent is the most widely used method of analyzing total phenolic content (TPC) in food extracts, and it correlates very well with the quantification results obtained via the HPLC-MS/MS technique. It happens, however, that the results of TPC obtained via this method may be overestimated due to the limitations in relation to samples containing a low concentration of phenolic compounds with a high content of ascorbic acid, dehydroascorbic acid or reducing sugars, which are common interference compounds [19].

Figure 1 shows the levels of TPC in pear pomace powders. Depending on the extraction method and particle size, TPC values ranged from 375.9 mg gallic acid (GA)/100 g dry weight (DW) (maceration granulation 75 µm) to 512.9 mg gallic acid/100 g DW (two-step extraction, granulation 710 µm) showing statistically significant differences between fractions with different particle sizes (*p* ≤ 0.05) and between extraction methods (*p* ≤ 0.05). Higher TPC amounts were generally obtained for fractions subjected to two-step extraction, from 0.5 up to 19.2% (for granulation 710 µm), compared to the samples extracted via maceration. This is probably due to the removal from the plant matrix of hydrophobic substances (including lipids, fatty acids, terpenoid compounds, etc.), which enhanced the subsequent extraction of phenolics (mostly hydrophilic) with an aqueous solution of methanol, assisted via ultrasound. So far, there are no data in the literature regarding proposed methods of extracting pomace flour. When the impact of particle size was analyzed, it seems that it has practically no influence on TPC (*p* ≤ 0.05) for either extraction method tested. For comparison, Wang et al. obtained lower TPC values for pears grown in Australia (depending on the variety) in the range of 189–314 mg GA/100 g DW, which is an almost 2-fold increase compared to 375–512 mg GA/100 g DW in pear pomaces [20]. This must be carefully compared, since Wang et al. were working with fresh fruit, and this work is in pomace; considering that around 65% of juice was extracted, this seems to have similar values [20]. For apple pomace subjected to a multi-stage extraction process (water, methanol and acetone), Reis et al. (2012) also obtained lower TPC values (around 256.6 mg GA/100 g DW) [19]. It is worth mentioning that differences in phenolic compounds content are due to fruit variety and harvest region (climatic and soil conditions) and also depend on the extraction method (solvent, time, sample-to-solvent ratio). In addition, seeds and petioles are also present in pomace, which may increase the content of phenolic compounds.

Flavonoids are important natural bioactive compounds, and their quantification in plant matrices can be performed with a quick and universal method using an aluminum chloride colorimetric assay against a flavonoid standard, most often quercetin. Pear pomace flour fractions with smaller particles showed lower total flavonoid content (TFC) (24.7–28.3 mg quercetin (QE)/100 g DW and 27.1–29.4 mg QE/100 g DW, from two-step extraction and maceration, respectively). For the fractions with bigger particles (710 µm and 180 µm), the highest TFC was found, around 30 mg QE/100 g DW. This can be explained by the larger-particle fractions’ inclusion of seeds and petioles, always richer in polyphenols. Again, regardless of particle size, samples that were subject to two-step extraction showed, on average, higher TFC (by 11%) when compared to the samples subject to maceration, since the first extraction method is more efficient, as explained earlier. For comparison, Wang et al. found almost 3 times higher TFC in fresh pear fruits of five cultivars, at the level of 57–153 mg QE/100 g DW [20]. This can be explained by the larger-particle fractions’ inclusion of seeds and petioles. Again, the comparison is not direct, as we are working with dry pomace flour. Rana et al. (2015) found a higher concentration of flavonoids in the range from 91 to 200 mg QE/100 g DW for apple pomace dried with different methods [21]. It should be remembered that flavonoids are labile compounds, easily degraded at higher temperatures and oxygen and light levels; therefore, the drying method could be of key importance for the obtained TFC, explaining this difference in results.

### 3.2. Antioxidant Activity: FRAP and DPPH

Fruits and vegetables in the human diet are one of the most important sources of antioxidants, showing a wide range of health-promoting properties [8]. Currently, there are many methods for assessing the antioxidant capacity of foods, based on different mechanisms of action and with specific limitations. Therefore, it is important to use, if possible, more than one method of measuring antioxidant activity in order to obtain comprehensive results [22].

In this study, the authors used the FRAP and DPPH methods as standard quantitative methods, both being simple, precise, sensitive and inexpensive and giving fast and reproducible results. FRAP is nonspecific, and any compound with a suitable redox potential will drive Fe^3+^-TPTZ reduction. On the other hand, the DPPH method depends on the solvent used [23].

Figure 2 shows the results of the antioxidant activity of the pear pomace flour, and it is worth noting that the antioxidant potential measured via FRAP assay was higher for fractions with larger particles (1–180 μm), regardless the extraction method used. However, a highest antioxidant potential was found in 710 and 75 µm fractions subjected to the two-step extraction method. The higher values of FRAP for larger particles are expected in accordance with the presence of seed and petioles in these fractions, already higher in polyphenols (Figure 1) For comparison, in studies on selected varieties of pears, Kolniak-Ostek et al. (2016) obtained similar values in the range of 109.38–391.45 mg Trolox/100 g DW [8].

The antioxidant potential measured via DPPH assay does not seem to be influenced by pear pomace flour particles being smaller than 1 mm, regardless of the extraction method used. However, it is considerably higher when using the two-step extraction method (an average increase of 38%). The highest antioxidant potential, as expected, was found in the fraction with the largest particles, at 1 mm (339.98 mg Trolox/100 g DW). It seems that, as in the case of TPC and TFC, the use of a two-step extraction method that uses hexane for removing non-polar compounds from the matrix contributed to a more efficient extraction of phenolics and other hydrophilic substances, which directly modulated the analyzed antioxidant activity.

To grasp the implications of antioxidant capacity concerning particle size, we need to consider two crucial facets: (i) the influence of mechanical forces on phytochemicals during the milling process, which may or may not result in cell disruption; and (ii) the influence of particle size in solid–liquid extractions. A pivotal element in achieving a more efficient extraction of bioactive components lies in the process of cell disruption, which can be achieved via various methods, including chemical, enzymatic, physical and mechanical processes. Reducing the particle size to submicron levels induces cell fragmentation, which in turn increases the specific surface area of the material, facilitating the extraction of valuable biological components contained within [24]. However, in some studies, reducing particle size did not always contribute to larger quantities of bioactive compounds retrieved. In addition, in some cases, granulometric characteristics did not influence the quantity of bioactive compounds extracted, which is the case presented in our study [24].

For comparison, in the studies by Kolniak-Ostek et al. (2016) [8], for selected fresh pear fruit cultivars, the values of DPPH antioxidant activity were almost twice as low, in the range of 99.62–192.72 mg Trolox/100 g DW [7]. On the other hand, in studies on different anatomical parts of pears, it was found that DPPH antioxidant activity increases in the following order: flesh < skin < seeds—106.62, 302.88 and 408.65 mg Trolox/100 g DW, respectively [8].

### 3.3. Antidiabetic Activity

A therapeutic strategy for addressing postprandial hyperglycemia, a critical early metabolic disturbance in type 2 diabetes, involves delaying digestion and reducing the absorption of intestinal glucose by inhibiting carbohydrate-hydrolyzing enzymes. Slowing the action of pancreatic α-amylase and intestinal α-glucosidase using natural plant inhibitors is a well-known and effective approach to managing type 2 diabetes.

Pre-treatment with hexane (Soxhlet extraction) provided an average of 1.5 times higher activity of pear pomace powder (Table 1) toward both enzymes involved in saccharide metabolism compared to one-step extraction with MeOH. Additionally, methanol cannot be used for food purposes. There was no linear relationship between antidiabetic activity and particle size. In this context, the smallest particles provided the highest α-amylase inhibition activity by pear pomace powder and sizes of 710 μm and again 53 μm the β-glucosidase inhibition activity.

Previous studies have identified pear skin as a more potent inhibitor of α-glucosidase activity than pear flesh, which has been linked directly to the skin’s high content of active phenolic compounds: chlorogenic acid, vanillic acid, ferulic acid and rutin; and triterpenes: oleanolic acid and ursolic acid [20,25,26]. Singh et al. (2021) also indicated that chlorogenic acid plays an important role in inhibiting α-amylase and α-glucosidase and shows hepatoprotective and antiatherosclerosis effects in streptozotocin-induced diabetic rats [27]. However, in this study, Pearson’s correlation analysis revealed negative correlations between the inhibitory activity of both hydrolases and TPC, TFC, FRAP (Table 2). Literature data provide information primarily on the strong inhibition of α-glucosidase as a key enzyme in carbohydrate digestion [26]. In turn, extraction of Pingguoli pear (*Pyrus pyrifolia*) fermentation broth revealed an ethyl acetate fraction rich in chlorogenic acid, caffeic acid and isoliquiritigenin (chalcone) with significantly stronger α-amylase and α-glucosidase activity than the aqueous fraction [28].

Thus, two-step extraction not only provided selective and more efficient extraction of compounds with hypoglycemic activity but also purified the matrix and eliminated compounds showing interference with the in vitro method used, such as some minerals, organic acids and amino acids. The limiting factor remains the compromise of using methanol as a relatively environmentally safe solvent and a medium that is efficient for polyphenols but at the same time interferes with the enzymatic activity of the extracts.

Although inhibition of carbohydrate breakdown is recognized as a key mechanism for regulating postprandial glycemia, other factors affecting the antidiabetic activity of pear pomace powder should also be considered. In addition to the benefits of the synergistic action of compounds of purified pomace powder extracts such as polyphenolic compounds, carotenoids and triterpenes with high antioxidant and anti-inflammatory activity directly linked to antidiabetic activity, pear pomace is a good source of dietary fiber, protein and magnesium (unpublished results).

Considering the above, pear pomace powder may be an attractive, competitive and natural by-product which is effective in delaying the effects of postprandial hyperglycemia while providing safe benefits without the undesirable side effects of antidiabetic drugs.

### 3.4. Antihypertensive Analysis as ACE (Angiotensin-Converting Enzyme) Inhibition Test

The most common comorbidity and complication of type 2 diabetes is hypertension. Hence, a complementary strategy to control enzyme activity may be to inhibit the activity of ACE (angiotensin-converting enzyme) involved in controlling blood pressure [29]. This process works by inhibiting ACE, which converts angiotensin I (AT I) into angiotensin II (AT II). As a result, the level of ATII in the blood decreases, which in excess constricts blood vessels, causing an increase in blood pressure.

Pear pomace powder proved to be an ACE inhibitor, with up to 68% inhibition potential for the sample quantity of 6.01 mg for 1 mL of enzyme (Table 1) and the smallest particles (53 µm). As with previous activities, extracts purified via the Soxhlet method inhibited ACE activity more strongly than those extracted with MeOH (*p* ≤ 0.05). In the case of two-step extraction, small particle size (75 μm and 53 μm) promoted high antihypertensive activity, which was not found for extracts treated with MeOH alone. Tukey’s multiple comparison test revealed pear pomace powder granulations of 180 μm and 75 μm to be the most favorable in terms of anti-ACE activity on MeOH. These results indicate particle size dependence for the two-step extraction.

Although the antihypertensive activity of the fruit is generally considered low to medium, of the 13 commonly consumed fruits, it was the pear that showed the highest ACE inhibitory potential, i.e., almost 5 times higher than for apple and 9 times higher than for pineapple and orange [30]. A recent study by Johnson et al. (2016) indicated that a 12-week intake of fresh pear can improve blood pressure and vascular function, particularly lowering systolic blood pressure and pulse pressure, in middle-aged men and women with metabolic syndrome [30]. Contrary to the studies of Ankolekar et al. (2012) on fermented pear juice, a correlation study indicated a negative relationship between TPC and ACE inhibitory activity (r = −0.44 and −0.74 for extraction with MeOH and two-step extraction [29].

Thus, pear pomace powder has the potential to provide dietary support as an ACE inhibitor effective in the treatment of high blood pressure, heart failure, type 2 diabetes and diabetic nephropathy. The multidirectional effect of pear pomace powder is also supported by the strong correlation between α-amylase inhibition activity and ACE inhibition activity (r > 0.78) (Table 2).

### 3.5. Antibacterial Activity

The antimicrobial properties of fruit phenolic compounds have been widely recognized for many years due to the presence of many active phytochemicals such as polyphenols, terpenoids, carotenoids, saponins and polypeptides, among others. Therefore, fruits present, to some extent, a high potential as antimicrobial agents and have even been suggested as a source of antibiotic alternatives [31]. As these compounds usually appear in very small amounts in fruits, they can lead to the occurrence of microbial spoilage [31]. In fact, in the case of pear, previous studies have shown that fresh fruits showed no antibacterial activity against any laboratory isolates, such as *Escherichia coli, Pseudomonas* spp., *Salmonella* spp. and *Bacillus* spp. [31]. However, the presence of large amounts of phenolic compounds registered in the pomaces allows us to infer that with the proper processing, there could be an enhancement of antibacterial activities. In this context, the potential inhibitory activity of pear pomace extracts on the growth of two model bacterial strains, *E. coli and S. aureus*, was determined using the pear pomace powdered samples at the same concentration of 5.0 mg/mL. The obtained results are presented in Figure 3 and are expressed as percentage of controls (bacteria growth with no pear extract added).

Results evidenced a noticeable higher inhibitory effect on *E. coli* than on *S. aureus*. It has been recognized that *S. aureus* is a particular species that is most often resistant to antibiotics [32], which make these results expected. Overall, bacteria growth inhibition was observed for both Gram-positive and Gram-negative bacterial strains, but most noticeably in the two-step extraction samples. In fact, for all the different particle size samples, this extraction yielded growth reductions higher than 60% in both bacteria species, suggesting a prominent antibacterial potential for extracts from this processing method. Since previous reports have shown that fresh pear extracts do not present antibacterial activity, the results corroborate the hypothesis that pear pomace processing can be a feasible way to enhance the fruit’s bioactivities.

Interestingly, treatments with MeOH only provided some antibacterial activities against *E.coli* in the higher granulometry samples (around 40% reduction).

Owing to their unique structural characteristics, Gram-negative bacteria exhibit higher resistance compared to Gram-positive bacteria, primarily due to the fact that the antibacterial effectiveness of most polyphenols relies on interactions with the surface of bacterial cells [32,33]. Consequently, the ability to detect antibacterial activity against Gram-negative bacteria holds substantial significance. In light of the escalating concern surrounding antimicrobial resistance, which is one of the most pressing global challenges, these findings hold promise for applications in both the food industry and the realm of nutraceuticals.

### 3.6. Agglomerative Hierarchical Clustering (AHC) and Principal Component Analysis (PCA)

AHC and PCA analysis examined the relationship between the TPC, biological activity and antibacterial activity of pear pomace powder (Figure 4). The AHC dendogram was based on Euclidean distance dissimilarity (in the 0–50 range) using Ward’s agglomeration method (Figure 4A). The dendrogram illustrates the hierarchical structure of the sample set due to the increasing diversity between them. The horizontal lines show the scale of similarity between the combined samples. An early combination of samples 2B-3B, 4B-5B and 2A-3A indicated the greatest similarity of these samples in terms of the analyzed composition and biological activity. Successive connections are separated by an increasing distance, which means a large difference between merged clusters. The dashed line in the graph indicates automatic pruning, leading to the formation of two homogeneous clusters: (1) pear pomace powder after MeOH extraction and (2) pear pomace powder after a two-step extraction. Within Cluster 2, a subgroup of pear pomace powders with larger particles was identified, which was also confirmed via PCA biplot. Two principal components (F1—60.88% and F2—23.19%) were identified, which explained 84.08% of the total data variance (Figure 4B). The other principal components were found to have no significant effect on the model. Longer vectors for antibacterial activities indicate a greater contribution by these primary variables to the construction of the components. On the basis of the small angles between the vectors representing the variables, strong positive correlations were found: (1) between activity toward *E. coli* and *S. aureus* and DPPH; (2) between FRAP, TFC and TPC and (3) between activity toward ACE and α-amylase. FRAP and β-glucosidase inhibition activity were not correlated, as the vectors of these variables are perpendicular. In contrast, no strongly negatively correlated variables were detected. PCA revealed that Cluster 1 had a lower content of phenolic compounds and health-promoting activity, in contrast to Cluster 2, where pomace powders of higher granularity 1B, 2B and 3B were distinguished by their high content of TFC, TPC and FRAP activity and pomace powders 4B and 5B with significantly higher biological activity toward inhibition of enzymes involved in sugar metabolism, ACE key in blood pressure regulation and antibacterial activity. Of those analyzed, samples 1B, 2B and 3B were the most similar to each other, as the distances of the points are small and differed the most with sample 4A. Perpendicularly located samples 1A, 2A, 3A and 5A with negative first component F1 and positive second component F2 indicated the lowest enzyme inhibition activities. Poor dietary habits and unhealthy lifestyles can cause high blood pressure, high blood glucose levels and increased oxidative stress. Hypertension is most often a coexisting disorder with diabetes, which is due to vascular stiffening and thus higher blood flow pressure, as well as obesity and the secretion and action of hormones. In addition, oxidative stress is another important cause and thus therapeutic target of diabetes. Therefore, pear pomace-based products, thanks to their antihypertensive (ACE inhibition), antidiabetic (α-amylase) and antioxidant effects (FRAP) correlated with TFC and TPC, will primarily have a strong synergistic and comprehensive preventive effect against the increase in cardiovascular risk. These beneficial health-promoting effects have also previously been attributed to the action of flavonoids and phenolic acids, so there is a significant need for a wider range of products that meet the goals of metabolic syndrome prevention [33,34].

## 4. Conclusions

The study confirmed that particle size of pear pomace powder and the method of extraction pre-treatment significantly modulate biological activity. Pomace powder after two-step extraction is characterized by a higher content of phenolic compounds, including flavonoids and higher antioxidant, antidiabetic, antihypertensive and antibacterial activity. Thus, two-step extraction provides sample cleaning from non-polar compounds and considerably improves the extraction. Moreover, the use of larger-particle pear pomace powders (710 µm, 180 µm and 1 mm) yielded TPC and TFC at higher concentrations, which directly resulted in high antioxidant activity (FRAP). The lower-granulation pear pomace powder variants (75 and 53 µm) were more effective in antidiabetic (as α-amylase and β-glucosidase inhibition), antihypertensive (as ACE inhibition) and antimicrobial (toward *E. coli* and *S. aureus*) activities. The results obtained provide the first database in the literature on the potential use of pear pomace in accordance with the concept of sustainable development, which represents a potential for future in vivo research. Pear pomace can be used as an ingredient in superfoods and functional foods, as well as an intermediate in cosmetic production and the pharmaceutical industry.

## Figures and Tables

**Figure 1 foods-12-04325-f001:**
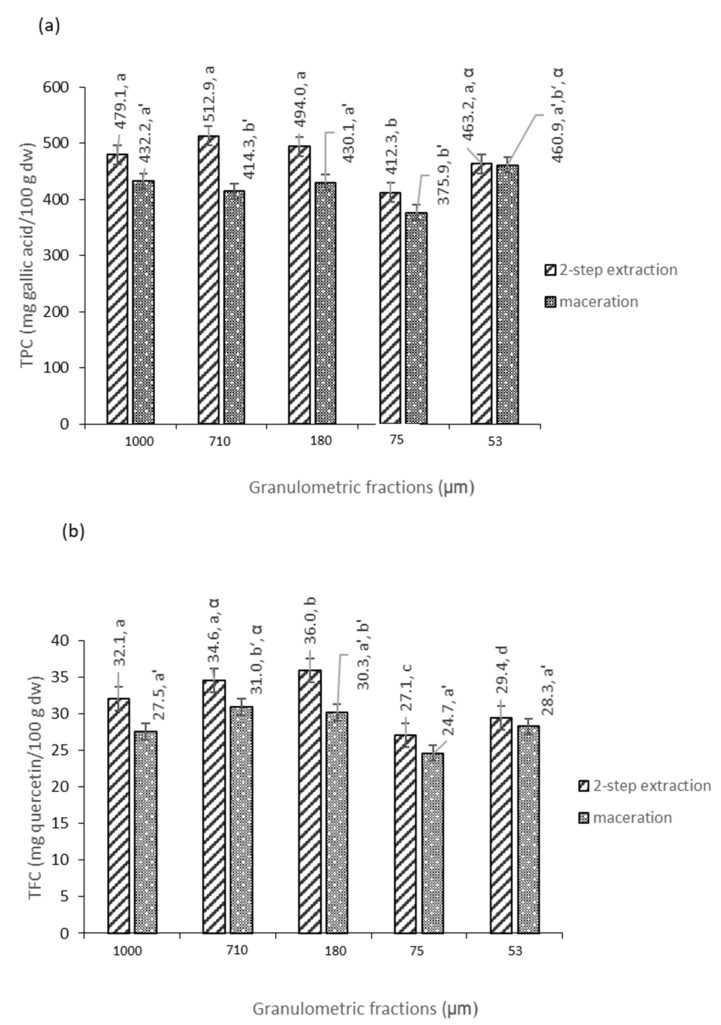
Total phenolic content (TPC) (**a**) and total flavonoid content TFC (**b**) of pear pomace flour. The data shown are mean values (n = 3) followed by an alphabet letter (for comparison between the different granulometric fractions subject to a two-step extraction), an alphabet letter and apostrophe (for comparison between the different granulometric fractions subject to maceration) or a Greek letter (when comparing extraction methods). Different letters mean significantly different results (*p* ≤ 0.05). DW: dry weight.

**Figure 2 foods-12-04325-f002:**
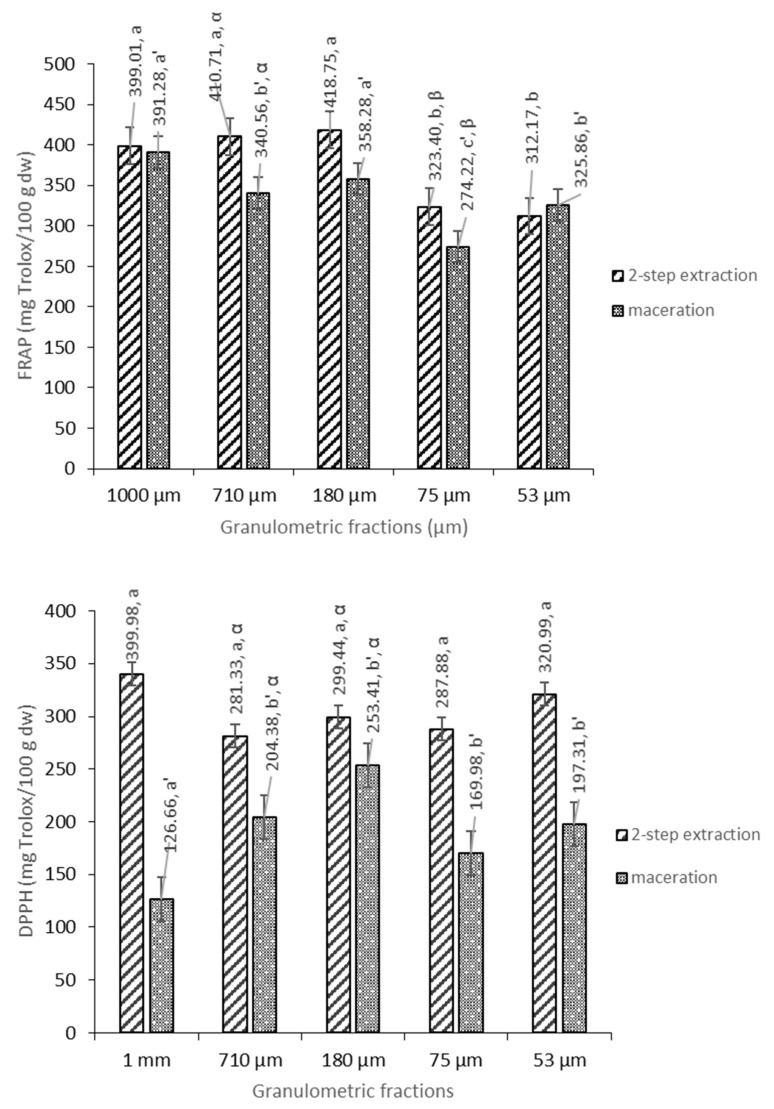
Antioxidant capacity of pear pomace flour measured via FRAP and DPPH assays. The data shown are mean values (n = 3) followed by an alphabet letter (for comparison between the different granulometric fractions subject to a two-step extraction), an alphabet letter and apostrophe (for comparison between the different granulometric fractions subject to maceration) or a Greek letter (when comparing extraction methods). Different letters mean significantly different results (Tukey’s HSD; *p* ≤ 0.05).

**Figure 3 foods-12-04325-f003:**
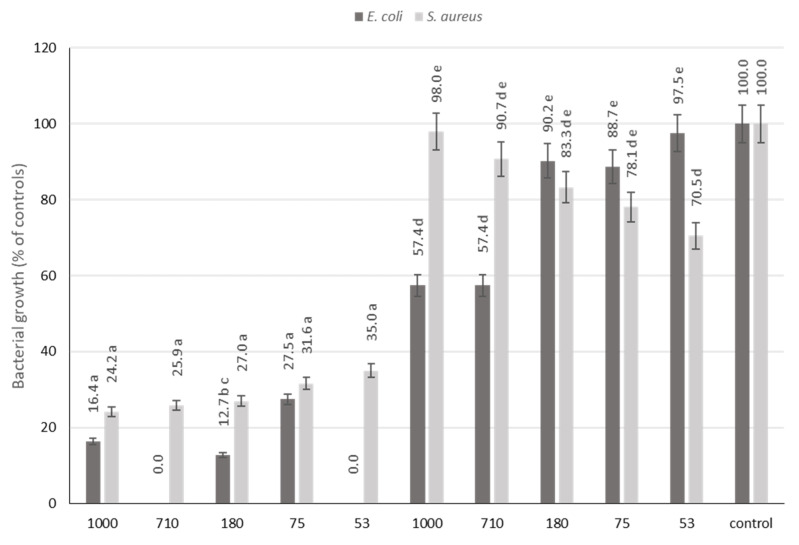
*E. coli* and *S. aureus* growth in the presence pear pomace powder at the same concentration of 5.0 mg/mL. The data shown are mean values (n = 3) followed by an alphabet letter (for comparison between the different granulometric fractions subject to a two-step extraction), an alphabet letter and apostrophe (for comparison between the different granulometric fractions subject to maceration). Different letters mean significantly different results (Tukey’s HSD; *p* ≤ 0.05).

**Figure 4 foods-12-04325-f004:**
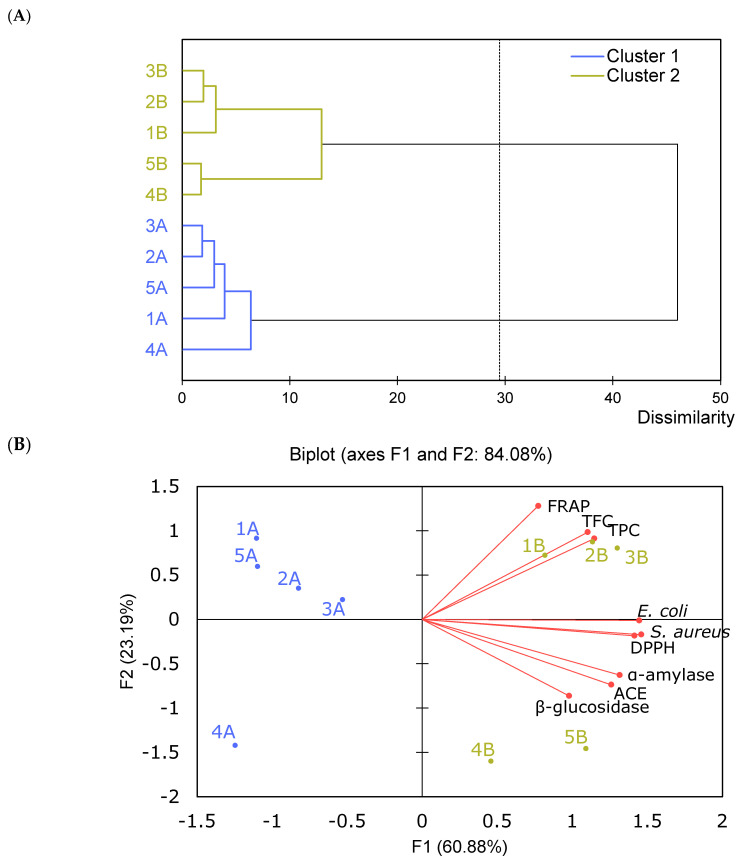
Agglomerative hierarchical clustering (AHC) (**A**) and principal component analysis (PCA) (**B**) of phenolic compounds, biological activity and antibacterial activity of pear pomace flour. Active observations: 1A–5A—maceration (cluster 2—blue); 1B–5B—two-step extraction (cluster 2—green). Active variables: TPC—total phenolic content; TFC—total flavonoid content; FRAP—ferric reducing antioxidant power; DPPH –2,2-diphenyl-1-picryl-hydrazyl-hydrate; ACE—angiotensin-converting enzyme. Variables “α-amylase”, “β-glucosidase” and “ACE” mean inhibitory activity against those enzymes (according to Table 1). Variables of “*E. coli*” and “*S. aureus*” mean antibacterial activity against them (according to Figure 3).

**Table 1 foods-12-04325-t001:** Antidiabetic and antihypertensive activity of pear pomace flour. The data shown are mean values (n = 3) followed by an alphabet letter (for comparison between the different granulometric fractions subject to a two-step extraction), an alphabet letter and apostrophe (for comparison between the different granulometric fractions subject to maceration. Different letters mean significantly different results (Tukey’s HSD; *p* ≤ 0.05).

Pear Pomace Flour			Antidiabetic Activity		Antihypertensive Activity
Extraction Method	Granulation (Mesh Size)	No.	α-Amylase Inhibition (%)	β-Glucosidase Inhibition (%)	ACE Inhibition (%)
two-step extraction	1000 µm	**1**	61.94 ± 1.16 a	7.00 ± 0.35 a	58.53 ± 0.97 a
	710 µm	**2**	57.44 ± 0.95 b	15.10 ± 0.66 b	55.94 ± 0.96 b
	180 µm	**3**	77.31 ± 0.92 c	11.14 ± 0.48 c	60.66 ± 1.26 c
	75 µm	**4**	68.77 ± 0.84 a	14.26 ± 0.44 d	65.38 ± 1.39 d
	53 µm	**5**	79.72 ± 1.03 c	15.40 ± 0.52 e	67.96 ± 1.12 e
maceration	1000 µm	**1**	46.79 ± 0.73 a’	4.21 ± 0.54 a’	48.89 ± 0.91 a’
	710 µm	**2**	46.06 ± 0.82 a’	9.37 ± 0.60 b’	43.97 ± 0.84 b’
	180 µm	**3**	53.27± 0.90 b’	7.57 ± 0.41 c’	55.35 ± 0.90 c’
	75 µm	**4**	53.30 ± 0.85 a’,b’	10.59 ± 0.58 d’	50.62 ± 1.09 d’
	53 µm	**5**	35.84 ± 0.79 c’	8.30 ± 0.46 e’	40.43 ± 0.83 e’

**Table 2 foods-12-04325-t002:** Pearson’s correlation coefficients (r) between phenolic compounds and biological activity of pear pomace powder.

Pearson Correlation Coefficients (r)									
	TPC	TFC	FRAP	DPPH	α-Amylase Inhibition	β-Glucosidase Inhibition	ACE Inhibition	*E. coli* Inhibition	*S. aureus* Inhibition
Extraction method: two-step extraction									
**TPC** (mg gallic acid/100 g DW)	1								
**TFC** (mg quercetin/100 g DW)	0.92	1							
**FRAP** (mg Trolox/100 g DW)	0.80	0.92	1						
**DPPH** (mg Trolox/100 g DW)	0.01	−0.10	−0.07	1					
**α-amylase inhibition** (%)	−0.29	−0.17	−0.45	0.17	1				
**β-glucosidase inhibition** (%)	−0.17	−0.29	−0.51	−0.67	0.21	1			
**ACE inhibition** (%)	−0.74	−0.75	−0.91	0.14	0.78	0.41	1		
***E. coli* inhibition** (%)	0.71	0.44	0.14	0.02	0.04	0.38	−0.16	1	
***S. aureus* inhibition** (%)	0.61	0.69	0.92	0.07	−0.69	−0.68	−0.94	−0.08	1
Extraction method: maceration									
**TPC** (mg gallic acid/100 g DW)	1								
**TFC** (mg quercetin/100 g DW)	0.50	1							
**FRAP** (mg Trolox/100 g DW)	0.59	0.56	1						
**DPPH** (mg Trolox/100 g DW)	0.17	0.61	−0.11	1					
**α-amylase inhibition** (%)	−0.75	−0.18	−0.12	0.14	1				
**β-glucosidase inhibition** (%)	−0.53	−0.16	−0.90	0.41	0.14	1			
**ACE inhibition** (%)	−0.44	−0.11	0.11	0.23	0.91	−0.13	1		
***E. coli* inhibition** (%)	−0.13	0.29	0.55	−0.48	0.09	−0.42	−0.08	1	
***S. aureus* inhibition** (%)	0.13	−0.26	−0.70	0.41	−0.33	0.61	−0.27	−0.92	1

## Data Availability

Data available within the article.

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
