# Peer review of "Influence of Particle Size and Extraction Methods on Phenolic Content and Biological Activities of Pear Pomace"

_foods, 2023, doi:10.3390/foods12234325_

Round 1

Reviewer 1 Report

Comments and Suggestions for Authors

Dear Authors 

COMMENTS- foods-2694893

·         The manuscript has the ambitious goal of providing an overview of the importance pear juice by-products as good source of phenolic content.

·         While I congratulate the author’s efforts, I do believe that there is still a scope for some revision of the manuscript.

1.      Research Title: Over all good and informative, require minor corrections, as below -

The phenolic content and biological activity of pear pomace influence of particle size

Influence of particle size and extraction methods on phenolic content and biological activities of pear (Pyrus comunis L.) pomace

2. Abstract: However the abstract part written inappropriately, also require many corrections and up-gradation for convey right massage to readers as per experimental results.  Rewrite this section-

Correction required as per copy enclosed-

3. Introduction: Required major editing on language up-gradation as well as incorporation problem, management, researches them, research hypothesis. Sufficient scientific information with inadequately offered for readers.  This section require redraft as per suggestion in enclosed PDF

4. Materials and Methods: This part has been nicely presented with detail and scientific methodology. No need for further corrections

5. Results: Satisfactorily presented with scientific facts, this portion also required summarization and minor modification for proper explanation of the experimental results. The tabular and graphical presentation also supports the results for easy to understand by readers. This section also required some minor correction.      

6. Discussion: Experimental results were extremely supported with closely related studies; it is very informative and good for all related scientific community, this part of manuscript required small changes and relocation at end of experimental results of each parameter with results.

7. Conclusion: Required revision with concluding remark of the authors.

8. Reference:  Adequately presented, required minor corrections and advise to follow the journal format for citations and references.

Comments on the Quality of English Language

Require minor editing of English language 

Author Response

Dear Referee, first of all, the authors would like to thank your valuable contribution to the overall improvement of the submitted manuscript.

Now, concerning you comments to the manuscript, we will answer point-by-point in the attach document.

Reviewer 2 Report

Comments and Suggestions for Authors

The valorisation of pear pomace through fractionation and subsequent application of organic solvent extractions shows the interesting bioactive potential of this by-product. The abstract clearly shows the objectives, the materials and methods as well as the results and the main conclusion. The study is very interesting and very well done and shown in the article.  However, some details should be improved before publication:

The valorization of by-products is essential in the agri-food industry, but we have to look for methods that are as sustainable as possible, why use a solvent such as methanol, which cannot be used for food purposes, instead of ethanol?

The application of ultrasound has been shown in other phenol-rich by-products as a technology that causes temperature rises in some points of the material that could degrade phenols, has this phenomenon been observed in this work?

Line 142. Centripetal force should be expressed in g and not in rpm.

Figure 1 and 2 isare outside the range of the sheet and there is a part that is not visible. On the other hand, it is not very clear what difference there is between a and a' or b and b', and it seems that between a and b there are significant differences. Please clarify further.

It is also not clear whether a 75 μm grind or a second one is best, or whether this grain size is easily scalable at industrial level or only used for analysis.

In figure 1 and 2 the data on the X-axis does not seem to correspond to the actual particle size of each sample. Change 1, 2, 3, 4 and 5 to 1000 μm, 710 μm, 180 μm, 75 μm, 53 μm.

The quality of the figures should be improved, as well as their compression, as there is a lot of data and they do not make the content of the figure clear.

In Table 1 change 1mm to 1000 µm to homogenize units. On the other hand, the Greek letters do not appear and it is not specified what the letters with apostrophe such as a' stand for.

In Table 1 change 1mm to 1000 µm to homogenize units. On the other hand, the Greek letters do not appear and it is not specified what the letters with apostrophe such as a' stand for.

Line 457 add that in addition methanol cannot be used for food purposes.

The activity results are quite good, but imply the use of very severe grindings. Are these grindings really feasible at industrial level? Would the high operating costs of grinding justify the products obtained?

Figure 3 change 1 mm per 1000 m, here if the size units are put right. The Greek letters detailed in the figure caption do not appear.

Figure 4 appears out of the picture and there is a part that cannot be seen.

The conclusions are clear but the critical point of view is missing, i.e. what is the hard approach that the results allow and what needs to be done further. Another very important aspect is the practical character of this work, although it presents very high antioxidant, antidiabetic, antihypertensive and antibacterial activities, how it would be done at industrial level, and if it would be the same using a solvent admitted in food such as ethanol instead of methanol.

Author Response

(The authors gave the same response as above.)

Reviewer 3 Report

Comments and Suggestions for Authors

In the presented manuscript, the authors focused on the influence of the particle size and the used extraction method on the content of phenols and flavonoids, as well as on the antioxidant, antidiabetic, antihypersensitivity and antibacterial effects of the investigated pear pomace. The idea of the work has a significant contribution to the existing knowledge about the bioactive properties of pear pomace, the analysis is meaningful and the results are adequately presented and clearly interpreted, but the following comments and suggestions should be considered:

Abstract: Remove subheadings like background methods and so on.

Introduction: In the last paragraph additionally emphasize the novelty of this paper.

Line 129, Where the pear pomace powder was obtained from and which variety of pear it is?

Line 310, 313, 316 When stating the authors of the paper and the year, for example Whang et al. 2021, put the year in parentheses because it is stated that way in the majority of the text, make it uniform throughout the entire paper.

Line 314, in is excess.

Line 342, Didn't the Rana et al. 2015 in their paper examine apple pomace?

Line 348-353,  Support these claims with references.

Line 395- 400, Give more examples for comparison of antioxidant activities, if not for pear pomace than for pomace of other fruits.

Line 460-470, Although this is written in the form of the author's conclusions, it is necessary to support the facts with references.

Line 527, 521, 538 Write  E.coli and S.aureus in italic.

Try to better explain how higher particle sizes affect the higher content of phenolic flavonoids and antioxidant values, while lower particle sizes show a greater effect on antidiabetic, antihypertensive and antimicrobial activities.

Technical improvement of paper is needed, first of all center the figures 1, 2 and 4 so that they become more visible.

Author Response

(The authors gave the same response as above.)

Reviewer 4 Report

Comments and Suggestions for Authors

Dear Authors, while trying to review your article, I noticed that some of the results are invisible (Figures 1, 2 and 4), which prevents me from verifying the manuscript. Please prepare the file correctly and resubmit for review.

At the same time, I am adding a few comments.

Latin names are written in italics, not in regular. 

In the descriptions of the methodologies, the authors indicate that they were made with modifications. Please indicate exactly what these modifications are.

Line 88, invalid citation style 

Centrifugation data, please convert rpm to g force. The rpm itself without the full specification of the centrifuge means nothing.

Authors sometimes write R2, other times r2. Why?

I will check the part regarding the results and discussion only when I see the correct charts.

Comments on the Quality of English Language

Quality of English is OK 

Author Response

(The authors gave the same response as above.)
